# SARS-CoV-2 Infection in Patients on Dialysis: Incidence and Outcomes in the Lazio Region, Italy

**DOI:** 10.3390/jcm10245818

**Published:** 2021-12-13

**Authors:** Claudia Marino, Laura Angelici, Valentina Pistolesi, Santo Morabito, Anteo Di Napoli, Enrico Calandrini, Silvia Cascini, Anna Maria Bargagli, Nicola Petrosillo, Nera Agabiti, Marina Davoli

**Affiliations:** 1Department of Epidemiology of the Regional Health Service—Lazio, Via Cristoforo Colombo, 112, 00147 Rome, Italy; c.marino@deplazio.it (C.M.); e.calandrini@deplazio.it (E.C.); s.cascini@deplazio.it (S.C.); a.bargagli@deplazio.it (A.M.B.); n.agabiti@deplazio.it (N.A.); m.davoli@deplazio.it (M.D.); 2Department of Internal Medicine and Medical Specialties, Hemodialysis Unit, Umberto I, Policlinico di Roma, “Sapienza” University School of Medicine, 00161 Rome, Italy; valentina.pistolesi@uniroma1.it (V.P.); santo.morabito@uniroma1.it (S.M.); 3National Institute for Health, Migration and Poverty, 00153 Rome, Italy; anteo.dinapoli@inmp.it; 4Infectious Disease and Infection Control Unit, Campus Bio-Medico, Medicine University Hospital, 00128 Rome, Italy; n.petrosillo@unicampus.it

**Keywords:** hemodialysis, end-stage kidney disease, SARS-CoV-2 infection, COVID-19, incidence, mortality, outcome

## Abstract

Patients with end-stage kidney disease represent a frail population and might be at higher risk of SARS-CoV-2 infection. The Lazio Regional Dialysis and Transplant Registry collected information on dialysis patients with a positive swab. The study investigated incidence of SARS-CoV-2 infection, mortality and their potential associated factors in patients undergoing maintenance hemodialysis (MHD) in the Lazio region. **Method:** The occurrence of infection was assessed among MHD patients included in the RRDTL from 1 March to 30 November 2020. The adjusted cumulative incidence of infection and mortality risk within 30 days of infection onset were estimated. Logistic and Cox regression models were applied to identify factors associated with infection and mortality, respectively. **Results:** The MHD cohort counted 4942 patients; 256 (5.2%) had COVID-19. The adjusted cumulative incidence was 5.1%. Factors associated with infection included: being born abroad, educational level, cystic renal disease/familial nephropathy, vascular disease and being treated in a dialysis center located in Local Health Authority (LHA) Rome 2. Among infected patients, 59 (23.0%) died within 30 days; the adjusted mortality risk was 21.0%. Factors associated with 30-day mortality included: age, malnutrition and fever at the time of swab. **Conclusions**: Factors associated with infection seem to reflect socioeconomic conditions. Factors associated with mortality, in addition to age, are related to clinical characteristics and symptoms at the time of swab.

## 1. Introduction

A novel coronavirus named severe acute respiratory syndrome coronavirus 2 (SARS-CoV-2), causing coronavirus disease 2019 (COVID-19), appeared in late 2019 in Wuhan, China, and rapidly spread progressively to other countries [1,2,3], with the WHO declaring it a pandemic and public health emergency of international concern in March 2020 [4]. As of 31 December 2020, 190 countries had confirmed cases, with Italy ranking eighth with 2,107,166 cumulative cases and 74,159 deaths [5]. According to the latest published data, in Rome, 5750 deaths in individuals positive to the SARS-CoV-2 occurred between 27 January 2020 and 6 April 2021 [6].

COVID-19 primarily manifests itself as an acute upper and lower respiratory tract illness that may be complicated by interstitial and alveolar pneumonia. It may also affect multiple other tissues such as the heart, digestive tract, kidneys, blood and nervous system [7,8,9]. According to the published literature, older patients with underlying chronic conditions such as diabetes mellitus, hypertension and cardiovascular disease tend to be more susceptible to COVID-19 and become severely ill [10]. To date, COVID-19 has caused a significant increase in the number of hospitalizations and intensive care unit admissions, with now well-investigated pulmonary, cardiac, vascular and renal complications [11,12,13,14,15].

The emergence of this pandemic put an additional strain on vulnerable population subgroups, such as end-stage chronic kidney disease (ESKD) patients, especially those on maintenance hemodialysis (MHD) [16].

MHD patients are at increased risk for COVID-19 infection and its complications for several reasons, primarily due to their dysregulated immune system because of their uremic state [17]. This is often accompanied by significant comorbidities such as cardiovascular disease, obstructive pulmonary disease, diabetes mellitus, obesity and cerebrovascular disease [18,19]. The logistical aspects of MHD treatment further increase the risk of SARS-CoV-2 infection; indeed, they require caregiver assistance and transportation from home to the dialysis unit and stay with dozens of patients undergoing MHD in a dialysis unit for several hours at a time, which may lead to widespread cross-contamination [20]. Hence, the management of MHD patients in the context of the pandemic presents several challenges and, in response, several guidelines aimed at mitigating the risk of infection spread in outpatient hemodialysis facilities have been already published [21,22,23,24,25].

Despite recent improvements in life expectancy among patients with ESKD, with a 28% decline in mortality over the last 16 years, the ESKD population has a higher mortality rate compared with the general population even after adjusting for age, race and diabetes mellitus [26]. With COVID-19, studies, including case reports or the experience of dialysis centers, suggest a more severe course of the disease and an increased risk of death in patients with chronic kidney disease, especially among those undergoing MHD [16,27,28,29,30,31,32,33,34,35,36,37].

To date, few studies have been conducted using ad hoc data on SARS-CoV-2 infection among MHD patients retrieved from dialysis registries [38,39].

The Lazio Regional Dialysis and Transplant Registry (RRDTL) was extended to include a new section on SARS-CoV-2 infection among patients undergoing dialysis treatment in the Lazio region. Information was retrieved through a questionnaire that collected information and monitored the health status of patients related to COVID-19 [40,41].

The aim of our study was to investigate the health effects of SARS-CoV-2 infection among MHD patients in the Lazio region, specifically considering incidence of SARS-CoV-2 infection, mortality within 30 days of infection onset and factors associated with the two outcomes in MHD patients.

## 2. Materials and Methods

### 2.1. Data Sources

The Lazio Regional Dialysis and Transplant Registry (RRDTL) is a population-based registry established in 1994 that collects detailed information on all patients undergoing chronic dialysis (e.g., those undergoing either hemodialysis or peritoneal dialysis for a period of at least 90 days). Information on sociodemographic status, clinical characteristics, dialysis treatments and drug therapy is collected for patients treated in all public and private accredited dialysis centers of the Lazio region in Italy. All dialysis units of the Lazio region are requested, by law, to register the information on their patients and to update the information every 6 months. If the patient no longer receives dialysis at the centers, the dialysis unit is required to communicate the date of termination of the dialysis treatment and the reason for the termination (kidney transplant, renal recovery, transfer to another dialysis center, death). [42] Details on the RRDTL are reported elsewhere [43,44,45,46,47]. In 2019, the RRDTL collected information on around 5000 dialysis patients [42].

A specific section, named “RRDTL COVID-19 section”, collecting information on SARS-CoV-2 infection, was included in the RRDTL, starting from March 2020. Data were collected for all dialysis patients who had a positive nasopharyngeal swab specimen to detect SARS-CoV-2 either in case of onset of symptoms or contact with a suspected or infected case of SARS-CoV-2 according to regional/national guidelines [21,22,23,24,25]. Information on symptoms on the day of infection (date of the first positive swab), need for hospitalization, complications and the end of infection (the date of the negative swab or the date of death), was collected.

The Lazio regional mortality registry was used to confirm the date of death registered in the RRDTL [48].

Moreover, to assess the data accuracy of the “RRDTL COVID-19 section”, MHD patients included in the study were linked through a record linkage procedure with an anonymous coding system with the regional “Emergenza CoronaVirus” platform for the study period (1 March–30 November 2020). In cases where a patient was found to have a positive swab only in the regional platform, the nephrologist in charge of the patient was contacted to clarify the patient’s status in terms of SARS-CoV-2 infection. If the infection status was confirmed, the nephrologist was asked to fill in the “RRDTL COVID-19 section”. The study population of infected patients was comprised of those that had the “RRDTL COVID-19 section” compiled [49].

### 2.2. Study Design and Population

A population-based prospective cohort study was performed. The cohort included all MHD patients resident in the Lazio region (over 5,700,000 residents), aged 18 years and over, treated between 1 March and 31 October 2020.

Cases between 1 March and 30 November 2020 were included in the study to assess infection status, and each infected patient was followed for 30 days from the infection onset date (date of first positive swab) to investigate 30-day mortality.

The outcomes investigated were: (1) SARS-CoV-2 infection between 1 March and 30 November 2020; (2) 30-day mortality among the infected patients.

### 2.3. Covariates

The following sociodemographic, clinical and treatment variables were recorded and tested as potential factors associated with the incidence of SARS-CoV-2 infection and 30-day mortality after infection: gender, age (18–64, 65–84, ≥85 years old), place of birth (abroad, Italy), educational level (illiterate, from 1–8 years of study, ≥9 years), self-sufficiency (total autonomy, autonomy in some activities, non-self-sufficient), type of nephropathy (glomerulonephritis, interstitial and toxic nephritis/pyelonephritis, cystic renal disease or familial nephropathy, renal malformation, renal vascular disease, diabetic nephropathy, systemic disease, unknown, other nephropathies; yes, no), comorbidities (severe hypertension, heart disease, diabetes, vascular disease, cerebrovascular disease, chronic pulmonary disease (COPD), cancer, thyroid disease, lipid metabolism’s alteration, motor deficit, liver disease, extra-uremic anemia, chronic inflammatory bowel disease (IBD), peptic ulcer, dementia, psychiatric disease, malnutrition, obesity, renal transplant; yes, no), local health authority (LHA) of dialysis center (named: Roma 1, Roma 2, Roma 3, Roma 4, Roma 5, Roma 6, Viterbo, Rieti, Latina and Frosinone), hours per week of dialysis (from 3–9, ≥10), hemodialysis vintage (one year or less, ≥2 years).

For patients with SARS-CoV-2 who died within 30 days from the infection onset, the following variables were also tested: infection time onset (March-July, August-November), symptoms at the time of swab (fever, cough, colds, conjunctivitis, respiratory distress, loss of taste and smell, gastrointestinal disease; yes, no), hospitalization (yes, no).

### 2.4. Statistical Analysis

To investigate the incidence of the SASR-CoV-2, infection variables were displayed according to infection status and χ2 test was used to test statistical significance of differences between subgroups (*p*-value < 0.05).

Univariate and multivariable log-binomial regression models were performed to estimate the crude and the age- and gender-adjusted risk of SARS-CoV-2 infection along with their 95% CI.

A multivariable logistic regression model, with a stepwise selection for the variables associated at univariate level with SARS-CoV-2 infection, was fitted to identify the potential factors associated with infection status (odds ratio—OR; 95% confidence interval—95% CI).

To investigate the 30-day mortality of infection onset, all variables were displayed according to vital status within 30 days of infection onset and χ2 test was used to test statistical significance of differences between subgroups (*p*-value < 0.05).

Univariate and multivariable log-binomial regression models were performed to estimate the crude and the age- and gender-adjusted risk of 30-day mortality along with their 95% CI.

A Cox regression model with a stepwise selection for the variables associated at univariate level with 30-day mortality was fitted to identify potential factors associated with 30-day mortality (hazard ratio—HR; 95% CI).

All analyses were performed using SAS Version 9.4 (SAS Institute, Cary, NC, USA). This study was carried out in full compliance with the current privacy laws.

## 3. Results

### 3.1. Characteristics of the Study Population by Infection Status

The cohort included 4942 MHD patients: 65.5% male, with an average age of 68.5 years (STD: 14.2 years). Of these, 4927 had the regional anonymous identifier necessary to carry out the record linkage procedures with the regional platform to verify SARS-CoV-2 infection. The “RRDTL COVID-19 section” identified 222 patients with a positive SARS-CoV-2 swab; among them, one could not be searched in the platform since he did not have an anonymous code, 195 (88%) were also found in the regional platform and 26 (12%) were not. The regional platform identified a further 87 MHD patients with a positive nasopharyngeal swab, and the patients’ nephrologists confirmed SARS-CoV-2 infection for 60 (69%) of them. The “RRDTL COVID-19 section” was filled out for only 34 MHD patients. Finally, the molecular swab did not confirm the nasopharyngeal swab for 21 (24%) MHD patients, and nephrologists could not provide information for six (7%) patients (Figure 1).

### 3.2. Infected Patients Considered in the Study

Thus, a total of 256 patients (64.1% male, mean age 68.4 years (STD: 14.1 years)) were identified as having COVID-19. The characteristics of the patients are summarized in Table 1 according to infection status. Italian natives were 88.6%; 48.6% were resident in Rome, 40.2% had more than 8 years of education and 62.2% of MHD patients were totally autonomous.

The most common cause of ESKD was diabetic nephropathy (19.6%), while heart disease and diabetes were the comorbidities (34.6% and 27.9%, respectively) with the highest prevalence. Half the patients were followed by a dialysis unit located in a LHA of Rome. Most of the patients (61.6%) had been on dialysis for 2 years or more; 80% of patients underwent dialysis treatment for more than 10 h a week.

A statistically significant difference (*p*-value < 0.05) in the distribution of covariates between non-infected and infected MHD patients was observed. Italian natives were, respectively, 89% and 82% in non-infected and infected patients. Illiteracy was highest among the infected (6.6% vs. 3.9%, respectively). Having cystic renal disease or familial nephropathy was more frequent among the infected MHD patients (12.1% vs. 7.6%, respectively), while having vascular disease was less common among the infected. Finally, a statistically significant difference was also detected in the LHA of the dialysis centers among infected/non-infected patients (Table 1).

### 3.3. Risk of SARS-CoV-2 Infection among Hemodialysis Patients and Main Risk Factors

The crude risk of SARS-CoV-2 infection was 5.18% (95% CI 4.58–5.85), while the age- and gender-adjusted risk was 5.05% (95% CI 4.34–5.89).

Variables significantly associated with the outcome at the univariate level were as follows: place of birth, cystic renal disease or familial nephropathy, vascular disease and LHA of hemodialysis center (Table 1).

The multivariable logistic regression model showed that place of birth other than Italy (OR 1.58, 95% CI 1.11–2.23), being illiterate vs. ≥9 years of education (OR 1.99, 95% CI 1.13–3.50) and cystic renal disease or familial nephropathy (OR 1.74, 95% CI 1.17–2.59) had a statistically significant association with an increased risk of infection, while vascular disease (OR 0.61, 95% CI 0.40–0.94) and being treated in dialysis centers belonging to an LHA other than Roma 2 (ORs less than 1) proved to be protective (Figure 2). Moreover, age group and gender were not risk factors for SARS-CoV-2 infection among MHD patients.

### 3.4. 30-Day Mortality from Infection Onset and Main Risk Factors

Of the 256 SARS-CoV-2-infected MHD patients, 59 patients (23%) died within 30 days (64.4% males, mean age 74.4 years; STD: 11.7 years). The patient characteristics according to vital status are shown in Table 2.

In the period August–November 2020, 85.5% of infections were recorded. The most common symptoms at the time of swab were fever (55.1%), cough (32.4%) and respiratory distress (20.3%); 75.0% of MHD patients infected by SARS-CoV-2 were hospitalized.

Statistically significant differences (*p*-value < 0.05) by vital status in MHD patients were found for: period of infection incidence (March–July: 25.4% in dead vs. 11.2% in alive patients), age (≥85 years: 23.7% in dead and 9.1% in alive patients), having thyroid disorder, being malnourished and having fever or respiratory distress at the time of first swab were more frequent in dead then in alive patients. The prevalence of severe hypertension was not a mortality risk factor (8.5% vs. 21.8%, dead vs. alive, respectively). Statistically significant differences in the distribution of dialysis centers’ LHA between alive/dead patients were found (Table 2).

The crude risk of 30-day mortality was 23.1% (95% CI 17.9–29.8), while the age- and gender-adjusted risk was 21.0% (95% CI 15.0–29.6).

Factors associated with 30-day mortality were being 85 years old or more (HR 5.31, 95% CI 2.47–11.42 ≥ 85 vs. 18–64 years), being malnourished (HR 5.44, 95% CI 2.62–11.28) and having a fever at the time of swab (HR 2.95, 95% CI 1.65–5.27) (Figure 3). Protective factors were not identified.

## 4. Discussion

End-stage kidney disease patients on MHD represent a high-risk subgroup for SARS-CoV-2 infection due to the higher probability of cross-contamination in closed environments and to the impairment of both adaptive and innate immunity related to their uremic state [50,51]. Moreover, the presence of multiple comorbidities (e.g., hypertension, cardiovascular disease, diabetes mellitus) in this population may be associated with a higher risk of adverse outcomes [50,51].

Our population-based cohort study provides information on the impact of SARS-CoV-2 among MHD patients in a large Italian region during the year 2020. Both the risk of infection (5.1%) and of mortality (21.0%) are high, confirming the greater vulnerability of this population and a worse outcome if compared with that reported in the general population [52].

In a previous study performed on the same population, during the period 1 March–31 July 2020, it was observed that there was a mortality risk of 5.5% higher in the dialyzed population with COVID-19 compared with the uninfected dialyzed population [53].

Regarding the impact of COVID-19 on the MHD population, a national survey conducted by the Italian Society of Nephrology during the first wave of the COVID-19 pandemic (April 2020), which considered data from 80.4% of Italian nephrology centers, reported a proportion of 3.55% of MHD patients to be positive for SARS-CoV-2 [54]. Such prevalence is lower than that reported here, and the difference could derive from the heterogeneity in the intensity of pandemic spread by region in Italy [55]; indeed, the highest proportion of infected patients was reported in northern Italy, probably as a consequence of the rapid growth of infected cases in the general population of that area [16], while a much lower proportion was detected in southern Italy, resulting in a lower average incidence in the country. For example, in one of the first regions to be hit by the pandemic, with dramatically high incidence rates, the Brescia Renal COVID Task Force reported SARS-CoV-2 RNA positivity in 94 out of 643 MHD patients (15%) [16]. Furthermore, the higher incidence of SARS-CoV-2 infection among MHD patients compared to the general population has been confirmed in several countries, independently from the phase of the pandemic [56,57,58]. These findings could be largely explained by some peculiarities of MHD patients, such as contact with healthcare providers 2–3 times a week and potential issues in physical distancing during transportation to dialysis units, in the waiting room and during the treatment session. Furthermore, multiple comorbidities of MHD patients could require additional day-to-day assistance or support which could impede social distancing. Similar to that reported elsewhere [59], our results are suggestive of different incidence of SARS-CoV-2 infection in relation to education level and place of birth. It could be speculated that these findings reflect socioeconomic factors and living conditions which do not favor the compliance of social distancing measures in place [57,58]. Moreover, the finding that the incidence of infection was higher in MHD patients assisted in a particular area of Lazio region (LHA Roma 2) could be partially explained by the documented higher incidence of infection in the general population living in that area [60].

The mortality risk reported in our study is comparable to that reported elsewhere. Indeed, in relation to the heterogeneity of the cohorts of MHD patients included in studies performed in various countries during different phases of the pandemic, the mortality rate appears to range between 20% and 35% [33,54,56,59,61,62,63]. For example, in a multicenter French cohort, including patients on MHD from 11 dialysis centers, a COVID-19 prevalence of 5.5% and a mortality rate of 28% were reported among 122 patients with a follow-up of at least 28 days [58]. Another study conducted in the UK, using the UK Renal Registry, reported a slightly lower mortality rate of 20%, possibly due to the shorter follow-up time period considered: 14 days from SARS-CoV-2 positivity [52].

As reported in previous studies [51,52,63], we found that advanced age and patients’ frailty, mainly malnutrition status, were associated with a higher risk of mortality.

Strengths of this study are the large sample size and the integrated use of several regional databases. These tools are subjected to continuous quality controlling that allow us to have accurate and standardized data; the completeness of the “RRDTL COVID-19 section” confirms the high quality of data collection. On the other hand, a limitation of the study is the lack of information about potential confounding clinical parameters not included in the RRDTL (e.g., inter-current acute illness unrelated to COVID-19 during hospital stay), limiting our capability to exclude other residual bias. Finally, dialysis patients did not have regular nasopharyngeal swabs as a screening tool; hence, it is possible that a minimal number of asymptomatic SARS-CoV-2 infection patients were not diagnosed and were excluded from the study.

## 5. Conclusions

In conclusion, our registry-based dialysis cohort study, conducted in the Lazio region, confirms that MHD patients represent a vulnerable group at particularly high risk of SARS-CoV-2 infection and severe health outcomes. The risk of infection appears to be higher in the presence of conditions that make difficult the maintenance of adequate social distancing and seems to reflect socioeconomic status and the prevalence of infection in the territory where patients underwent dialysis treatment. It is therefore important to ensure the adherence to the recommendations of the guidelines implemented to mitigate the risk of infection spread in outpatient hemodialysis facilities. It is also necessary to ensure high vaccination coverage in these frail patients. Moreover, presence of fever at the time of swab emerged as a significant 30-day mortality risk factor, indicating the need for constant monitoring and additional care of this subgroup starting from the early phase of infection. The adoption of extended indications for hospitalization in COVID-19 MHD patients was recommended, especially in the presence of fever or need for oxygen therapy at the time of swab.

## Figures and Tables

**Figure 1 jcm-10-05818-f001:**
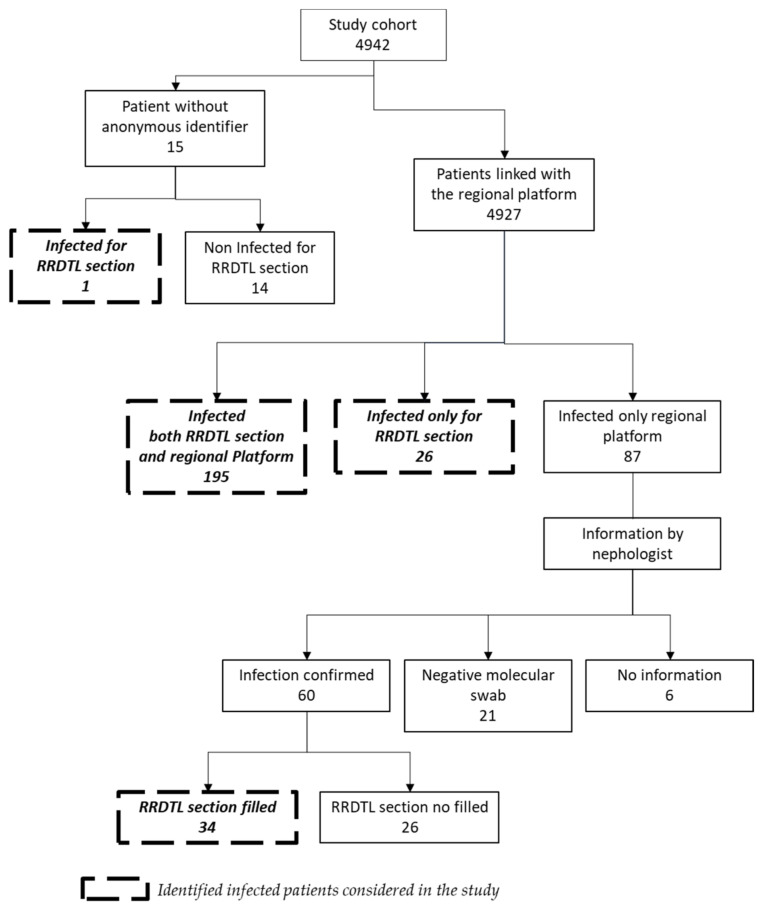
Flowchart to assess the completeness of the “RRDTL COVID-19 section”.

**Figure 2 jcm-10-05818-f002:**
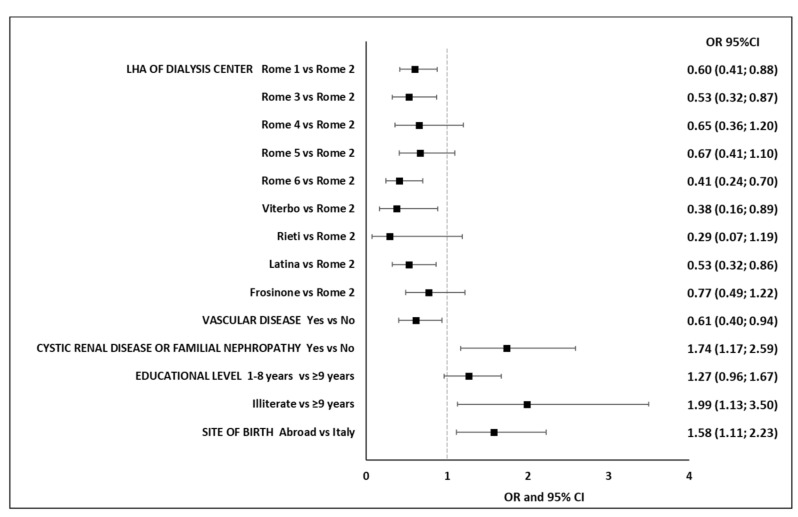
Factors associated with SARS-CoV-2 infection among hemodialysis patients.

**Figure 3 jcm-10-05818-f003:**
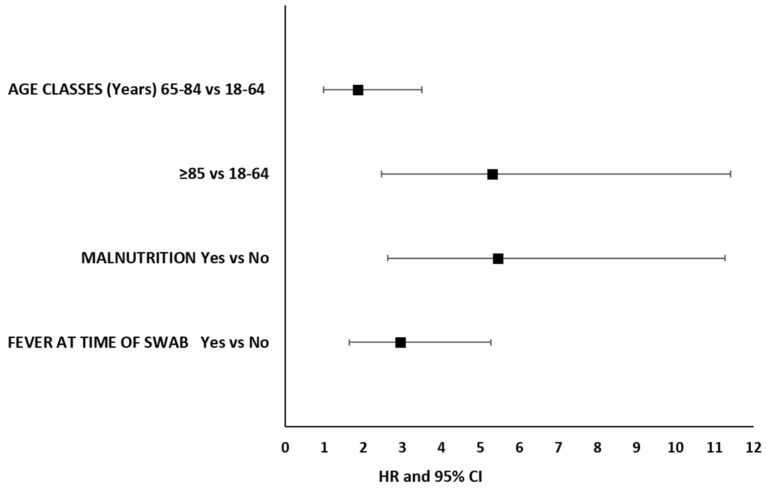
Factors associated with 30-day mortality in hemodialysis patients infected with SARS-CoV-2.

**Table 1 jcm-10-05818-t001:** Demographic and clinical characteristics of hemodialysis patients by infection status.

		Total		Non-Infected		Infected		*p*-Value χ2
		*N*	%	*N*	%	*N*	%	
**Total**		4942		4686		256		
**Gender**								0.609
	**Male**	3239	65.5	3075	65.6	164	64.1	
	**Female**	1703	34.5	1611	34.4	92	35.9	
**Age (years)**								0.843
	**18–64**	1710	34.6	1618	34.5	92	35.9	
	**65–84**	2713	54.9	2577	55.0	136	53.1	
	**≥85**	519	10.5	491	10.5	28	10.9	
**Site of birth**								0.001
	**Italy**	4379	88.6	4169	89.0	210	82.0	
	**Abroad**	563	11.4	517	11.0	46	18.0	
**Residence**								0.171
	**Rome municipality**	2403	48.6	2264	48.3	139	54.3	
	**Rome province**	1243	25.2	1187	25.3	56	21.9	
	**Other Lazio municipalities**	1296	26.2	1235	26.4	61	23.8	
**Educational level (years)**								0.049
	**Illiterate**	198	4.0	181	3.9	17	6.6	
	**1–8**	2757	55.8	2610	55.7	147	57.4	
	**≥9**	1987	40.2	1895	40.4	92	35.9	
**Self-sufficiency**								0.351
	**Total autonomy**	3073	62.2	2919	62.3	154	60.2	
	**Autonomy in some activities**	1118	22.6	1051	22.4	67	26.2	
	**Non self-sufficient**	751	15.2	716	15.3	35	13.7	
**Nephropathy**								
	**Diabetic nephropathy**	969	19.6	912	19.5	57	22.3	0.271
	**Renal vascular disease**	915	18.5	863	18.4	52	20.3	0.447
	**Glomerulonephritis**	452	9.1	430	9.2	22	8.6	0.753
	**Cystic renal disease or familial nephropathy**	389	7.9	358	7.6	31	12.1	0.010
	**Interstitial and toxic nephritis/pyelonephritis**	183	3.7	175	3.7	8	3.1	0.615
	**Systemic disease**	152	3.1	143	3.1	9	3.5	0.675
	**Renal malformation**	52	1.1	50	1.1	2	0.8	0.663
	**Other nephropathies**	621	12.6	596	12.7	25	9.8	0.165
	**Unknown**	1209	24.5	1159	24.7	50	19.5	0.059
**Comorbidity**								
	**Heart disease**	1711	34.6	1616	34.5	95	37.1	0.390
	**Diabetes**	1379	27.9	1307	27.9	72	28.1	0.935
	**Severe hypertension**	785	15.9	737	15.7	48	18.8	0.198
	**Obesity**	772	15.6	723	15.4	49	19.1	0.111
	**Vascular disease**	749	15.2	724	15.5	25	9.8	0.014
	**Cerebrovascular disease**	684	13.8	644	13.7	40	15.6	0.396
	**Cancer**	670	13.6	639	13.6	31	12.1	0.487
	**COPD**	609	12.3	573	12.2	36	14.1	0.385
	**Thyroid disease**	607	12.3	580	12.4	27	10.5	0.385
	**Lipid metabolism’s alteration**	447	9.0	427	9.1	20	7.8	0.480
	**Renal transplant**	400	8.1	381	8.1	19	7.4	0.686
	**Malnutrition**	293	5.9	275	5.9	18	7.0	0.443
	**Liver disease**	215	4.4	208	4.4	7	2.7	0.193
	**Motor deficit**	142	2.9	134	2.9	8	3.1	0.805
	**Extra-uremic anemia**	131	2.7	127	2.7	4	1.6	0.266
	**IBD**	129	2.6	119	2.5	10	3.9	0.182
	**Peptic ulcer**	90	1.8	86	1.8	4	1.6	0.751
	**Psychiatric disease**	89	1.8	81	1.7	8	3.1	0.102
	**Dementia**	83	1.7	75	1.6	8	3.1	0.065
**LHA of dialysis centre**								0.014
	**Rome 1**	967	19.6	920	19.6	47	18.4	
	**Rome 2**	1026	20.8	949	20.3	77	30.1	
	**Rome 3**	489	9.9	468	10.0	21	8.2	
	**Rome 4**	275	5.6	262	5.6	13	5.1	
	**Rome 5**	425	8.6	403	8.6	22	8.6	
	**Rome 6**	523	10.6	505	10.8	18	7.0	
	**Viterbo**	199	4.0	193	4.1	6	2.3	
	**Latina**	88	1.8	86	1.8	2	0.8	
	**Rieti**	501	10.1	479	10.2	22	8.6	
	**Frosinone**	449	9.1	421	9.0	28	10.9	
**Weekly hours of dialysis**								0.104
	**3–9**	1008	20.4	966	20.6	42	16.4	
	**≥10**	3934	79.6	3720	79.4	214	83.6	
**Hemodialysis vintage (years)**								0.214
	**0–1**	1900	38.4	1811	38.6	89	34.8	
	**≥2**	3042	61.6	2875	61.4	167	65.2	

**Table 2 jcm-10-05818-t002:** Characteristics of hemodialysis patients infected with SARS-CoV-2 by vital status within 30 days of infection onset.

		Total		Alive		Dead		*p*-Value χ2
		*N*	%	*N*	%	*N*	%	
**Total**		256		197		59		
**Period of infection incidence**								0.006
	**March-July**	37	14.5	22	11.2	15	25.4	
	**August-November**	219	85.5	175	88.8	44	74.6	
**Gender**								0.950
	**Male**	164	64.1	126	64.0	38	64.4	
	**Female**	92	35.9	71	36.0	21	35.6	
**Age (years)**								0.005
	**18–64**	90	35.2	76	38.6	14	23.7	
	**65–84**	134	52.3	103	52.3	31	52.5	
	**≥85**	32	12.5	18	9.1	14	23.7	
**Site of birth**								0.075
	**Italy**	210	82.0	157	79.7	53	89.8	
	**Abroad**	46	18.0	40	20.3	6	10.2	
**Residence**								0.114
	**Rome municipality**	139	54.3	100	50.8	39	66.1	
	**Rome province**	56	21.9	46	23.4	10	16.9	
	**Other Lazio municipalities**	61	23.8	51	25.9	10	16.9	
**Educational level (years)**								0.622
	**Illiterate**	17	6.6	14	7.1	3	5.1	
	**1–8**	147	57.4	110	55.8	37	62.7	
	**≥9**	92	35.9	73	37.1	19	32.2	
**Self-sufficiency**								0.705
	**Total autonomy**	154	60.2	120	60.9	34	57.6	
	**Autonomy in some activities**	67	26.2	52	26.4	15	25.4	
	**Non self-sufficient**	35	13.7	25	12.7	10	16.9	
**Nephropathy**								
	**Diabetic nephropathy**	57	22.3	48	24.4	9	15.3	0.140
	**Renal vascular disease**	52	20.3	38	19.3	14	23.7	0.457
	**Cystic renal disease or familial nephropathy**	31	12.1	22	11.2	9	15.3	0.399
	**Glomerulonephritis**	22	8.6	16	8.1	6	10.2	0.623
	**Systemic disease**	9	3.5	8	4.1	1	1.7	0.387
	**Interstitial and toxic nephritis/pyelonephritis**	8	3.1	4	2.0	4	6.8	0.066
	**Renal malformation**	2	0.8	1	0.5	1	1.7	0.364
	**Other nephropathies**	25	9.8	18	9.1	7	11.9	0.536
	**Unknown**	52	20.3	42	21.3	8	13.6	0.187
**Comorbidity**								
	**Heart disease**	95	37.1	72	36.5	23	39.0	0.734
	**Diabetes**	72	28.1	57	28.9	15	25.4	0.599
	**Obesity**	49	19.1	39	19.8	10	16.9	0.626
	**Severe hypertension**	48	18.8	43	21.8	5	8.5	0.021
	**Cerebrovascular disease**	40	15.6	27	13.7	13	22.0	0.122
	**COPD**	36	14.1	26	13.2	10	16.9	0.467
	**Cancer**	31	12.1	23	11.7	8	13.6	0.697
	**Thyroid disease**	27	10.5	16	8.1	11	18.6	0.021
	**Vascular disease**	25	9.8	16	8.1	9	15.3	0.106
	**Lipid metabolism’s alteration**	20	7.8	16	8.1	4	6.8	0.736
	**Renal transplant**	19	7.4	14	7.1	5	8.5	0.725
	**Malnutrition**	18	7.0	8	4.1	10	16.9	0.001
	**IBD**	10	3.9	6	3.0	4	6.8	0.194
	**Motor deficit**	8	3.1	5	2.5	3	5.1	0.324
	**Dementia**	8	3.1	5	2.5	3	5.1	0.324
	**Psychiatric disease**	8	3.1	6	3.0	2	3.4	0.894
	**Liver disease**	7	2.7	4	2.0	3	5.1	0.207
	**Extra-uremic anemia**	4	1.6	3	1.5	1	1.7	0.926
	**Peptic ulcer**	4	1.6	4	2.0	0	0.0	0.270
**LHA of dialysis centre**								0.033
	**Rome 1**	47	18.4	35	17.8	12	20.3	
	**Rome 2**	77	30.1	58	29.4	19	32.2	
	**Rome 3**	21	8.2	15	7.6	6	10.2	
	**Rome 4**	13	5.1	9	4.6	4	6.8	
	**Rome 5**	22	8.6	20	10.2	2	3.4	
	**Rome 6**	18	7.0	11	5.6	7	11.9	
	**Viterbo**	6	2.3	4	2.0	2	3.4	
	**Latina**	2	0.8	0	0.0	2	3.4	
	**Rieti**	22	8.6	19	9.6	3	5.1	
	**Frosinone**	28	10.9	26	13.2	2	3.4	
**Weekly hours of dialysis**								0.501
	**3–9**	42	16.4	34	17.3	8	13.6	
	**≥10**	214	83.6	163	82.7	51	86.4	
**Hemodialysis vintage (years)**								0.155
	**0–1**	80	31.3	66	33.5	14	23.7	
	**≥2**	176	68.8	131	66.5	45	76.3	
**Symptoms at time of swab**								
	**Fever**	141	55.1	100	50.8	41	69.5	0.011
	**Cough**	83	32.4	61	31.0	22	37.3	0.363
	**Respiratory distress**	52	20.3	34	17.3	18	30.5	0.027
	**Colds**	11	4.3	9	4.6	2	3.4	0.695
	**Gastrointestinal disease**	7	2.7	5	2.5	2	3.4	0.725
	**Conjunctivitis**	5	2.0	4	2.0	1	1.7	0.870
	**Loss of taste and smell**	2	0.8	2	1.0	0	0.0	0.437
**Hospitalization for infection**								0.549
	**No**	64	25.0	51	25.9	13	22.0	
	**Yes**	192	75.0	146	74.1	46	78.0	

## Data Availability

Data related to the findings reported in our manuscript are available to all interested researchers upon reasonable request and with the permission of the Regional Department because of stringent legal restrictions regarding the privacy policy on personal information in Italy (national legislative decree on privacy policy n. 196/30 June 2003). For these reasons our dataset cannot be made available on public data deposition.

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
