# Peer review of "SARS-CoV-2 Infection in Patients on Dialysis: Incidence and Outcomes in the Lazio Region, Italy"

_jcm, 2021, doi:10.3390/jcm10245818_

Round 1

Reviewer 1 Report

In the present article by Marino et al. with the title "SARS-CoV-2 infection in patients on dialysis: incidence and outcomes in the Lazio region, Italy", the authors provide a thorough statistical analysis of the factors associated with SARS-CoV-2 infection and mortality in patients on dialysis. While the paper would deserve to be considered for publication, several issues would need to be addressed to finalize it.

The subsection "Aim" seems inappropriate in the section "Materials and Methods" (lines 76-80).

The subsections "Follow-up" and "Outcomes" could be included in the subsection "Study design and population" (lines 117-123).

The subsection "Statistical analysis" is empty (line 144). "Characteristics of hemodialysis patients infected with SARS-CoV-2 by vital status within 30 days of infection onset" is likely the name for a subsection (lines 218-219).

The subsection "30. -day mortality from infection onset and main risk factors" is too short and superficial (lines 236-241). Are there any protective parameters?

The "Conclusion" is also incomplete (lines 309-318). Is there any practical advice for the scientists and/or doctors? For instance, the sentence on lines 295-298 is appropriate for the "Conclusion".

Reviewer 2 Report

The paper by Marino et al. describes the incidences, mortality and factors associated with SARS-CoV-2 infection of dialysis patients in the Lazio region, Italy.

The authors demonstrate that socioeconomic factors contribute to infection rates, while mortality is associated with age, clinical characteristics, and condition at the time of initial diagnosis of a SARS-CoV-2 infection. 

The study is interesting and should help to improve management of dialysis patients during the SARS-CoV-2 pandemics.

There are a few comments relating to the paper:

  1. Line 155: should read “30-day mortality...” instead of “30.-day...”
  2. Figure 1 is a bit confusing, the flow chart should be improved. For instance, it would be helpful if there was another box containing the 256 patients of this study and with arrows from which groups they came; optionally, different colours could be used for easier understanding.
  3. Figure 1: “plataform” should be corrected to “platform”
  4. The patient numbers in Figure 1 should be revised, there are some inconsistencies between Figure 1 and the text:
    • number in box “infected only for RRDTL section 26” should be 27 (as in the text line 173)
    • number in box “infected only regional platform” is divided into “infection confirmed = 54”, “negative molecular swab” = 21, “no information 6” = total number is 81 patients instead of 87?
    • infection confirmed 54 = 34 filled and 26 no-filled = 60 (the are six patients missing somewhere)
  5. Line 212: should read “Figure 2” instead of “Figure 1”.
  6. Figure captions should be below the figures
  7. Conclusions: A consequence of these findings could be added to the conclusion, what could be further improved to mitigate the risk for dialysis patients during this pandemics?
